 

ⓐ | **Open Peer Review** | Clinical Microbiology | Research Article

# Insights into respiratory microbiome composition and systemic inflammatory biomarkers of bronchiectasis patients

Aleksandras Konovalovas,[1,2] Julija Armalytė,[1] Laurita Klimkaitė,[1] Tomas Liveikis,[1] Brigita Jonaitytė,[3] Edvardas Danila,[3,4] Daiva Bironaitė,[2] Diana Mieliauskaitė,[2] Edvardas Bagdonas,[2] Rūta Aldonytė[2]

**ABSTRACT** The human microbiomes, including the ones present in the respiratory tract, are described and characterized in an increasing number of studies. However, the composition and the impact of the healthy and/or impaired microbiome on pulmonary health and its interaction with the host tissues remain enigmatic. In chronic airway diseases, bronchiectasis stands out as a progressive condition characterized by microbial colonization and infection. In this study, we aimed to investigate the microbiome of the lower airways and lungs of bronchiectasis patients together with their serum cytokine and chemokine content, and gain novel insights into the pathogenesis of bronchiectasis. The microbiome of 47 patients was analyzed by sequencing of full-length 16S rRNA gene using amplicon sequencing Oxford Nanopore technologies. Their serum inflammatory mediators content was quantified in parallel. Several divergently composed microbiome groups were identified and characterized, the majority of patients displayed one dominant bacterial species, whereas others had a more diverse microbiome. The analysis of systemic immune biomarkers revealed two distinct inflammatory response groups, i.e., low and high response groups, each associated with a specific array of clinical symptoms, microbial composition, and diversity. Moreover, we have identified some microbiome compositions associated with high inflammatory response, i.e., high levels of pro- and anti-inflammatory cytokines, whereas other microbiomes were in correlation with low inflammatory responses. Although bronchiectasis pathogenetic mechanisms remain to be elucidated, it is clear that addressing microbiome composition in the airways is a valuable resource not only for diagnosis but also for personalized disease management.

**IMPORTANCE** The population of microorganisms on/in the human body resides in distinct local microbiomes, including the respiratory microbiome. It remains unclear what defines a healthy and a diseased respiratory microbiome. We investigated the respiratory microbiome in chronic pulmonary infectious disease, i.e., bronchiectasis, and researched correlations between microbiome composition, systemic inflammatory biomarkers, and disease characteristics. The bronchoalveolar microbiome of 47 patients was sequenced, and their serum inflammatory mediators were quantified. The microbiomes were grouped based on their content and diversity. In addition, patients were also grouped into low- and high-response groups according to their inflammatory biomarkers' levels. Certain microbiome compositions, mainly single-species dominated, were associated with high levels of inflammatory cytokines, whereas others correlated with low inflammatory response and remained diverse. We conclude that respiratory microbiome composition is a valuable resource for the diagnostics and personalized management of bronchiectasis, which may include preserving microbiome diversity and introducing possible probiotics.

**Peer Reviewers** Sutonuka Bhar, University of Florida, Gainesville, Florida, USA; Danielle Elizabeth Campbell, Washington University in St Louis School of Medicine, St Louis, Missouri, USA; Evann Hilt, Department of Laboratory Medicine & Pathology, Minneapolis, Minnesota, USA

Address correspondence to Rūta Aldonytė, ruta.aldonyte@imcentras.lt.

The authors declare no conflict of interest.

See the funding table on p. 16.

**KEYWORDS** bronchiectasis, respiratory microbiome, lower respiratory tract microbiome, systemic inflammatory biomarkers

The importance of human microbiota, a full array of microorganisms that live on and inside the human organism's niches and modulate human health, has been highlighted by numerous studies since its discovery (1). Recently, microbiota science dramatically excelled due to the progress of emerging sequencing technologies, permitting compositional and functional assessments, and forever shifted our knowledge about the role of microorganisms in human health and disease (2). It is known that the human body is colonized by a diversity of bacteria, fungi, viruses, phages, protozoa, and archaea, all of which together participate in the bioconversion of nutrients, production of necessary metabolites, immune response maintenance, and resistance against pathogenic microbes (3). The broader studies where host/microorganism interactions are characterized using the -omics approach, e.g., metabolomics, exposomics, secretomics, and similar, are increasingly performed. Although the loss of protective and beneficial functions of microbiota leads to dysbiosis and subsequent dominance of certain microbes that cause various diseases (4), the exact interrelationships between microbiota changes and diseases' mechanisms remain enigmatic.

Depending on the residency, human microbiota can be classified into major groups: gut, oral, respiratory, vagina, and skin, with subgroups according to the precise location (1). The gut microbiota is the most studied and considered to have the most significant impact on human health through several crucial processes such as food fermentation, vitamin production, stimulation of immune response, neurotransmitter release, and many others (5, 6). The human gut microbiota possesses more than a thousand microbial species and has been characterized as a vital organ having a multidirectional connection with neural, endocrinal, immunological humoral, and metabolic pathways (7, 8). The second most abundant microbiota, after the gut, is the oral microbiota (9). It was suggested that certain oral microbiota shifts are closely related to the pathogenesis of systemic diseases like rheumatoid arthritis (10), nephritis (11), endocarditis (12), inflammatory bowel disease (13, 14), and cancer (15).

Similarly to other microbiotas, human respiratory microbiota constitutes a crucial layer of defense within the airways and lungs in addition to clearing, warming, and humidifying systems. Recent technology development and subsequent investigations increasingly underscore the importance of respiratory microbiota in healthy and pathological states within the respiratory tract (16). It was shown that bacterial burden in the upper respiratory tract could be up to 10,000-fold greater than in the lower respiratory tract due to the close contact with the ambient air and microorganism sources in the environment (17). The presence and composition of microorganisms in the lower respiratory tract are determined by their immigration from the upper respiratory tract (oral, nasal microbiota, or the environment), their elimination from the airways (by mucociliary clearance and cough), and their relative reproduction, which is decided by the conditions in the lungs (18). As the main function of the human respiratory system is the exchange of oxygen and carbon dioxide, it requires a large surface area, i.e., 40-fold larger than the human skin. Hence, the maintenance of the epithelial barrier is crucial and depends on tight junction molecules such as claudins, occludin-1, and others (19). Claudins are a family of proteins, which, along with occludin, are the most important components of the tight junctions, and their integrity might be reinforced by the strains of microbiota (20). Similarly, the epithelial barrier proteins have a crucial role in the regulation of the integrity of the airway and respiratory epithelium (21), and the integrating effects of local microbiota were well documented (22).

The microorganisms residing within the upper airway of bronchiectasis (BE) or chronic obstructive pulmonary disease (COPD) patients enter the lower levels of the tract and cause the exacerbation or progression of the disease (23). BE is a chronic lung disease with heterogeneous clinical conditions and outcomes (24). It is recognizable by specific radiological patterns, e.g., abnormal widening and thickening of its airway

wall, airway plugging, mosaic attenuation, and volume loss, and clinically—by chronic cough, sputum production, and recurrent respiratory infections. The persistent infectious process with infiltration of immune cells, particularly neutrophils, is a central pathophysiological event in BE.

The role of microbiota in BE pathogenesis remains poorly investigated in comparison to other respiratory diseases (25, 26). The broader investigation and understanding of microbiota shifts in BE will allow us to improve therapeutic strategies or find new therapeutic targets. It is becoming increasingly clear that it is possible to identify multiple BE endotypes based on comorbidities, underlying causes, and airways infectious agents involved. The endotype-defining biomarkers include microbiome data needed to guide the treatments and move toward personalized medicine for BE (27). Indeed, a recent study revealed distinct host/microbiome interaction patterns among neutrophilic *Haemophilus*-predominant, neutrophilic balanced microbiome, and eosinophilic subgroups in COPD and BE (28–30). Monitoring patients' microbiome variability and inflammatory status may provide valuable information for the diagnostics and management of BE patients.

Therefore, the main goal of this study was to investigate how the microbiomes of BE patients' lower respiratory tract correlate with the levels of circulating inflammation biomarkers and provide information on possible pathogenetic mechanisms of lung diseases in association with the shifts of microbiomes.

## MATERIALS AND METHODS

### Study design and patients' characteristics

This study has been approved by the Lithuanian Bioethics Committee (approval #2021/2-1308-786) in accordance with the current guidelines and regulations. Informed consent was obtained from all the participants. The specimens from the lower airways were collected from a diverse cohort of 47 individuals diagnosed with BE, allowing for an exploration into a wide array of disease manifestations and stages. Of these, 37 samples were bronchial aspirate, whereas the remaining 10 were bronchoalveolar lavage (BAL) fluid (Table S1, found at https://zenodo.org/doi/10.5281/zenodo.13305856). A detailed breakdown of the data is provided in Table 1, including an average number of antibiotic treatment episodes per year. To assess possible associations with the patient's health status, comprehensive health metrics were also collected: body mass index (BMI), forced expiratory volume (FEV), forced vital capacity (FVC), and the diffusing capacity of the lungs for carbon monoxide (DLCO). An in-depth analysis of these pivotal health metrics can be found in Table 2 (individual patient data are presented in Table S1).

### Fiberoptic bronchoscopy and sample collection

Fiberoptic bronchoscopy and BAL were performed as described elsewhere (31). The subjects were premedicated with atropine, and lidocaine was delivered topically via an atomizer. The bronchoscope was inserted transnasally (in most cases) or orally and passed to the segmental or subsegmental bronchus. In case of a sufficient amount of bronchial secretion, suction of mucus to the sterile container was carried out. If the amount of bronchial secretion was insufficient for the suction, BAL was performed in

**TABLE 1** Demographic and clinical characteristics of study participants with bronchiectasis[a]

| Patient characteristics (*n* = 47) | Patient number (%) | Average (SD) | Median (range) |
|---|---|---|---|
| Sex (female) | 31 (65.9%) | | |
| Age, years | | 64.9 (12.4) | 66 (31–86) |
| BMI, kg/m$^2$ | | 24.4 (4.0) | 24.0 (15–32) |
| Disease duration, years | | 8.6 (10.4) | 3.0 (1–30) |
| Antibiotic therapy episodes per year | | 1.1 (1.2) | 1 (0–6) |

[a]SD, standard deviation; BMI, body mass index. The range is expressed as minimum value to maximum value.

**TABLE 2** Respiratory functions of study participants[a]

| Sample type and respiratory functions | Patient number (%) | Average (SD) | Median (range) |
|---|---|---|---|
| Bronchial aspirate | 37 (78.7%) | | |
| Bronchoalveolar lavage fluid | 10 (21.3%) | | |
| FEV1% | 42 (89.4%) | 83.7 (26.9) | 87.5 (24–129) |
| FVC% | 41 (87.2%) | 101.2 (20.6) | 101 (62–154) |
| FEV1/FVC% (SD, sample number) | 42 (89.4%) | 70.4 (16.2) | 72 (23–109) |
| DLCO% | 25 (53.2%) | 77 (21.2) | 73 (40–120) |

[a]Respiratory function indices: FEV1%, forced expiratory volume in 1 second (predicted %); FVC, forced vital capacity (predicted %); DLCO, diffusing capacity of the lungs for carbon monoxide (predicted %). SD, standard deviation.

the right middle lobe, lingual, or in the area of greatest radiologic abnormality. Bronchial aspirate and BAL fluid (BALF) samples were used to analyze microbial composition.

In parallel, the patients' sera were collected to assess the systemic levels of major inflammatory mediators.

## DNA purification and sequencing

Bronchial aspirate and BALF samples were centrifuged at 12,000 × $g$ and 4°C for 15 minutes, and pellets were stored at −80°C until the DNA extraction step. DNA extraction was performed using Zymo Research Quick-DNA Microprep Plus Kit (Biological Fluids & Cells protocol) according to the manufacturer's recommendations.

The sequencing library for 16S rRNA was prepared using the SQK-RAB204 or SQK-16S024 rapid 16S amplicon barcoding kits (Oxford Nanopore Technologies [ONT]), adhering to the guidelines provided by the manufacturer. The entire 16S rRNA gene was amplified from each sample using 10 ng/uL of total DNA, following the protocols provided by Oxford Nanopore Technologies (available at https://community.nanopore-tech.com). This amplification was executed using LongAmp Taq polymerase master mix (New England Biolabs) and the ONT-supplied barcoded primer pair (27F and 1492R). The PCR process followed a specific temperature cycling program. It began with a 1-minute denaturation at 95°C, proceeded through 35 cycles: 20 seconds at 95°C, 30 seconds at 55°C, and 1 minute at 65°C, and concluded with a final extension phase lasting for 5 minutes at 65°C. According to ONT instructions, the barcoded amplicons were purified using AMPure XP beads (Beckman Coulter). Sample DNA concentration was determined using the NanoDrop One spectrophotometer (Thermo Fisher Scientific) and pooled up to 10 samples in equimolar concentrations for 100 pmol. The library was loaded onto the R9.4.1 Flongle flowcell and was sequenced for 24 hours using a MinION with Flongle adapter (ONT). All sequencing data are publicly available in the European Nucleotide Archive (https://www.ebi.ac.uk/ena) under accession number PRJEB70318.

## Full-length 16S rRNA sequencing analysis

The raw fast5 files underwent base-calling with Guppy (version 6.5.7+ca6d6af) (Guppy basecalling software, https://community.nanoporetech.com [2023]) and minimap2 (32) (version 2.24-r1122), producing fastq files. Default settings were maintained, except for the use of the dna_r9.4.1_450bps_sup model, which was chosen to enhance base-calling precision, and excluding reads with a mean q-score below 8. The reads in the fastq files, generated from the base-calling process, were demultiplexed, and the adapters were trimmed using Porechop (https://github.com/rrwick/Porechop) (version 0.2.4). The taxonomic annotation and relative abundance/count data for the basecalled and demultiplexed fastq files were determined using the Emu (33) software (version v3.4.4), employing its default settings and database for the analysis.

We conducted a cluster analysis of the lung microbiome based on species' relative abundance using the weighted UniFrac distance (34) method via the phyloseq (35) package (version 1.42.0). Then, we utilized hierarchical clustering with the Ward.D2 approach, as detailed by Murtagh and Legendre (36), and identified optimal cluster

numbers using silhouette and elbow score methods. We used the Kruskal-Wallis rank sum test to compare differences between multiple groups and performed pairwise group comparisons using the pairwise Wilcoxon rank sum test. We adjusted the *P* values of both tests using the Benjamini and Hochberg method. For visualization, we employed several R packages, including ggplot2 (37) (version 3.4.4), ggtree (38) (version 3.8.0), ggdendro (version 0.1.23), ggsignif (39) (version 0.6.4), and factoextra (version 1.0.7). All analyses were performed in R version 4.2.2.

## Analysis of inflammatory serum biomarkers

Twenty inflammation-associated proteins were quantified in serum samples by using a ProcartaPlex Human Inflammation 20-Plex (Thermo Fisher Scientific) multiplex immunoassay kit. A single dilution was used. The proteins analyzed were as follows: CD62E, CD62P, GM-CSF, ICAM-1, IFNα, IFNγ, IL-1α, IL-1β, IL-10, IL-12p70, IL-13, IL-17A, IL-4, IL-6, IL-8, IP-10, MCP-1, MIP-1α, MIP-1β, and TNFα. All the procedures were performed according to the manufacturer's protocol, and data were acquired on the Luminex 200 analyzer. Results, i.e., individual patient data, are presented in Table S3 (Table S3, found at https://zenodo.org/doi/10.5281/zenodo.13305856) and contain absolute and normalized values.

Inflammation-associated protein levels in serum samples were quantified using the 20-Plex Human ProcartaPlex Panel. Post-data acquisition, log transformation, was applied to normalize the results on a scale from 0 to 1, enhancing consistency and comparability. Protein targets that had detection rates below 60% were excluded from the analysis; this included IL-6 and IL-8. IL-10 was also excluded from the analysis because it showed a detectable signal in only four out of the 43 samples. Similarly, GM-CSF was excluded due to detectable signals in only three out of 43 samples.

To classify protein target profiles based on their role in inflammation, we employed k-means clustering. The elbow method and silhouette scores determined the optimal number of clusters.

We formed distinct groups reflecting different inflammatory response patterns based on these clustering results. These groups were determined by their respective inflammatory response clusters. For statistical comparison of protein differences across groups, we employed the Kruskal-Wallis rank sum test for multiple group comparisons and the pairwise Wilcoxon rank sum test for pairwise comparisons. *P* values from both tests were adjusted using the Benjamini and Hochberg method. All analyses were performed using R version 4.2.2.

## RESULTS

### Demographic and clinical characteristics of participants

In this study, the specimens from the lower airways were collected from a diverse cohort of 47 individuals diagnosed with BE, allowing for an exploration into a wide array of disease manifestations and stages. Of these, 37 were bronchial aspirate, and the remaining 10 were BALF samples, which are both sample types widely used for microbiome testing. A detailed breakdown of the patients' data is provided in Table 1, showcasing a broad demographic representation across genders and age groups: 16 males and 31 females with an average age of 64.9 years (SD = 12.4). A notable characteristic of the studied population was the substantial variation in disease duration, which averaged around 8 and a half years, underscoring the commitment to examine a wide variety of patients and disease progressions. The participants were further analyzed through the lens of the pulmonary function characteristics, revealing a spectrum of disease severity. The average antibiotic therapy episode number was 1.1 times per year (SD = 1.2), shedding light on the medical management of their condition through a therapeutic lens. To ensure a holistic understanding of the health status of our participants, we delved into comprehensive health metrics including BMI, FEV1, FVC, FEV1/FVC ratio, and DLCO. The pivotal health metrics can be found in Table 2.

## Bacterial community composition of BE patients' samples

The most common phyla in the lower respiratory tract of BE patients were *Proteobacteria* and *Firmicutes*, with relative abundance averages reaching 59% and 35%, respectively. All the patients had representatives of these two phyla, and the majority (>75% of the patients) also had bacteria belonging to phyla *Bacteroidetes* and *Fusobacteria* (with average relative abundances of 4% and 2%, respectively) in their lower respiratory tract microbiomes (Table S2, found at https://zenodo.org/doi/10.5281/zenodo.13305856). A low abundance (0.1%) of *Actinobacteria* was found in >50% of the patients (the average relative abundance for the patients who had *Actinobacteria* was 0.3%). From the *Proteobacteria* phylum, the families *Pasteurellaceae, Pseudomonadaceae,* and *Enterobacteriaceae* were found in high relative abundance, with *Pasteurellaceae* being the most common (98% of the patients' microbiotas contained an average relative abundance of 27%). *Pseudomonadaceae* and *Enterobacteriaceae* were found in less than 50% of the patients; however, their average relative abundance when present in the patients' microbiomes was high (34% and 23%, respectively). *Neisseriaceae* and *Campylobacteraceae* were also common *Proteobacteria* among the patients' microbiota, with lower relative abundance (<5%). In the *Firmicutes* phylum, the most common and abundant families were *Veillonellaceae* and *Streptococcaceae*, detected in the majority (96%) of the patients' microbiomes with an average relative abundance of 14% and 13%, respectively, whereas all the other families displayed a significantly lower relative abundance of 1% or less. *Prevotellaceae* family from the phylum *Bacteroidetes* was present in 78% of the patients, with an average relative abundance of 3%, whereas the other two families were of lower abundance. Phylum *Fusobacteria* was represented by two families, only one of which, *Fusobacteriaceae*, was present in 75% of the patients' microbiomes with a relative abundance of around 1%.

In total, 473 different bacterial species were identified by phylogenetic analysis; however, only 39 of them were observed in more than half of all the samples (Table S2). When analyzing the species belonging to *Proteobacteria*, several patterns could be observed: the bacteria known as respiratory tract pathogens (*Haemophilus influenzae, Pseudomonas aeruginosa, Klebsiella pneumoniae*) were found in less than a half of the patients' microbiomes (45%, 28.5%, and 18%, respectively). However, when detected in the samples, their relative abundance was very high: the relative abundance of *H. influenzae* in the BE patient samples was 35%; in some patients, it reached 99.8%. The relative abundance of *P. aeruginosa* was 36%, reaching up to 99%, whereas for *K. pneumoniae*, it was 21%, reaching up to 89%. Aside from the mentioned pathogens, other species belonging to the *Proteobacteria*, such as *H. parainfluenzae, Haemophilus haemolyticus, Neisseria subflava,* and *Campylobacter concisus*, could be found in more than 50% of the patients' microbiomes; however, their relative abundance was lower, reaching 2% to 8%. The most common *Firmicutes* genera were *Veillonella* (*V. dispar* and *V. parvula*) and *Streptococcus* (*S. mitis* and *S. oralis*). The species were present in over 85% of the patients' microbiomes, with a relative abundance of 2%–8%. However, several patients had a prominent increase in *V. dispar, V. parvula,* or *S. mitis* abundance, reaching up to 20%–38%. From phylum *Bacteroidetes*, the most common and abundant bacteria detected belonged to the genus *Prevotella*; however, their relative abundance was below 1%, and only in single samples an increase of up to 3% could be detected. Interestingly, one patient had a considerable increase of *Fusobacterium nucleatum* (phylum *Fusobacteria*), reaching up to 20% of relative abundance. However, in other lower respiratory tract samples, *Fusobacteria* species were found only in low abundance (below 0.5%).

In order to find similarities among the lower respiratory tract microbiotas, hierarchical clustering analysis was performed, revealing 15 clusters (Fig. 1). The clusters could be grouped into two cluster groups: clusters 1–8 containing samples with higher diversity, with the highest abundance of several species of *Firmicutes* phylum (further *Firmicutes* clusters) and higher Shannon diversity index (the highest Shannon diversity index of 3.8 in cluster 8; the lowest of 2.3 in cluster 5), and clusters 9–12 and 14–15 displaying significantly lower diversity (a Shannon diversity index below 1 in majority of samples). In

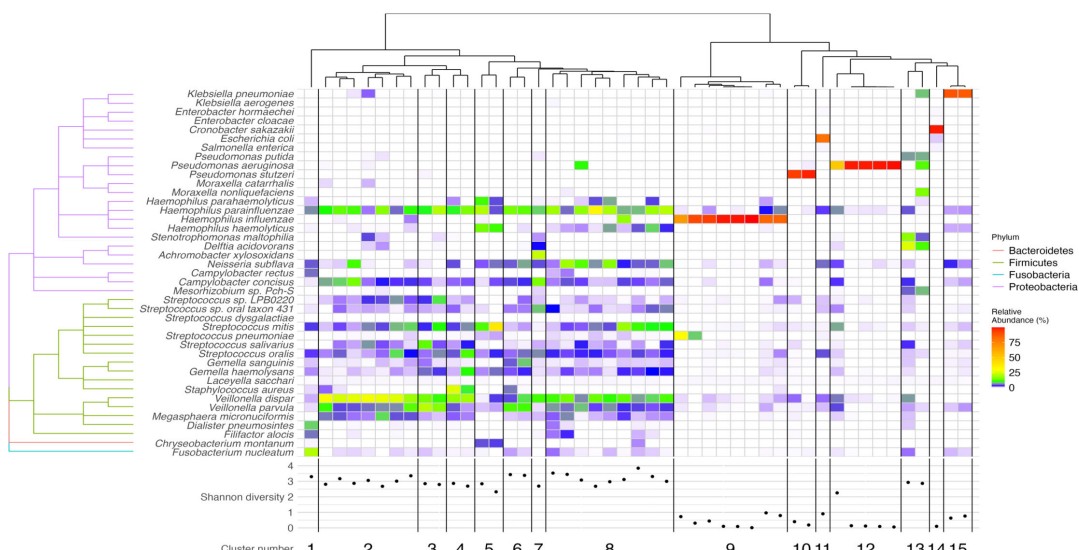

**FIG 1** Lung microbiome diversity in bronchiectasis patients. The heatmap illustrates the relative abundance of 41 bacterial species, derived by selecting the seven most abundant species from each sample across diverse samples and grouped into four distinct phyla: *Proteobacteria, Firmicutes, Bacteroidetes,* and *Fusobacteria*. Each row symbolizes a unique bacterial species, whereas each column indicates a different sample. Based on clustering indices, an optimal number of 15 clusters was identified, four of which comprised more than two study samples. Below, the Shannon diversity scatterplot depicts the species diversity within each sample.

the latter group, a single species of dominant bacteria was apparent (Fig. 1). Interestingly, in the grouped bacterial clusters with the low diversity, cluster 13 had a higher Shannon index of 2.9.

In the cluster group displaying a single dominant species (clusters 9–15), the dominant bacteria were more generally associated with the respiratory tract and opportunistic infections. The largest cluster in this group had dominant *H. influenzae,* whereas smaller clusters had dominant *Pseudomonas* spp. and bacteria belonging to the *Enterobacteriaceae* family. The eight patients belonging to cluster 9 with a dominant *H. influenzae* (further *H. influenzae*-dominant cluster) had a very high relative abundance of 67.5% to 99.8% of the single dominant species. Additionally, the presence of *H. influenzae* was previously detected by culture methods (data not shown). Interestingly, in the samples that were not grouped into *H. influenzae*-dominant cluster, a presence of *H. influenzae* could also be detected, although in significantly lower numbers (relative abundance mostly lower than 1%). One of the patients belonging to a higher diversity (*Firmicutes*; cluster 8) also had a comparatively high relative abundance (17%) of *H. influenzae*.

Eight other species of the *Haemophilus* genus could also be found in the samples, with the most common being *H. parainfluenzae* (43 patients, 91.5%), *H. haemolyticus* (23 patients, 49%), *Haemophilus* sp. oral taxon 036 (20 patients, 42.6%), and *H. parahaemolyticus* (14 patients, 30%). The most common and abundant was *H. parainfluenzae*, and it was observed in all the samples of the *Firmicutes* cluster group, where it was one of the most abundant bacterial species, with its relative abundance reaching up to 33%.

The second most common dominant species was *P. aeruginosa* (cluster 12 was the *P. aeruginosa*-dominant cluster). Four out of five patients' microbiomes with dominant *P. aeruginosa* displayed over 98% of *P. aeruginosa* relative abundance. Similarly to *H. influenzae*, the presence of dominant *P. aeruginosa* in the patients was also confirmed by culturing methods (data not shown).

Other species belonging to the *Pseudomonas* genus were also found (22 different species); however, 15 of them were detected only in single samples. Even with the presence of a high variety of species, *Pseudomonas* species was less commonly found than *Haemophilus* in the samples of BE patients: *P. aeruginosa* was present in 13 patients

(28%), *P. putida* in 8 patients (17%), and *P. stutzeri* in 7 patients (15%). A small cluster with another dominant *Pseudomonas* species, *P. stutzeri* (relative abundance over 93%), was also discerned (Fig. 1), although the cluster consisted of only two samples.

The remaining clusters displaying one dominant species had bacteria belonging to the *Enterobactereiaceae* family (*K. pneumoniae, Escherichia coli, Cronobacter sakazakii*); however, only one to two patients displayed this type of microbiome composition. Cluster 13, although grouped with other clusters that had one dominant species, displayed different characteristics. It had a higher Shannon diversity index, and the dominant bacterial species could not be distinguished. However, an increase in the relative abundance of several bacterial species related to respiratory tract infections could be observed in the samples of cluster 13, including *K. pneumoniae, P. aeruginosa*, or bacteria known to cause opportunistic infections (*Moraxella nonliquefaciens, Stenotrophomonas maltophilia, Delftia acidovorans*) (Fig. 1).

In the clusters displaying a higher diversity of microbiota (clusters 1–8), an abundance of bacteria typical for the healthy lung could be observed, with the highest relative abundance of *Firmicutes*. Because *Firmicutes* are a part of a healthy human lower respiratory tract microbiota (40), it was expected that bacteria belonging to the genera *Streptococcus* and *Veillonella* were found in nearly 96% of all patient microbiomes, not only in the samples belonging to the higher-diversity cluster group. *S. mitis* and *S. oralis* were the most abundant and common among the patients, with the relative abundance reaching up to 4% and 2%. *Streptococcus pneumoniae* was found in more than 63% of the patients (30 patients); however, the relative abundance of this respiratory tract pathogen rarely exceeded 1%. The genus *Veillonella*, *V. dispar* and *V. parvula*, was the most abundant and common among all the patients, with 8% and 3% average, whereas in some samples, the relative abundance reached up to 29% and 21%, respectively. *V. dispar* was the most abundant in the samples from *Firmicutes* cluster 2. However, even if the mentioned bacteria could be detected in the majority of the samples, their relative abundance was considerably higher in the high-diversity cluster group (Fig. 1).

## Distinct systemic inflammatory profiles of the bronchiectasis patients

From all 20 biomarkers in the ProcartaPlex Panel, which include CD62E, CD62P, ICAM-1, IFNα, IFNγ, IL-1α, IL-1β, IL-4, IL-6, IL-8, IL-10, IL-12, IL-13, IL-17A, IP-10, MCP-1, MIP-1α, MIP-1β, TNFα, and GM-CSF, we observed low detection rates for IL-6 and IL-8 in 38 and 27 samples, respectively. As these proteins were detected in less than 60% of the samples, they were excluded from further clustering analysis. IL-10 was also excluded from this part of the analysis because it showed a detectable signal in only four out of the 43 samples. Similarly, GM-CSF was excluded due to detectable signals in only three out of the 43 samples.

Firstly, we performed k-means clustering using values from 16 biomolecules. To determine the number of clusters in our data set, we employed the consensus elbow method and silhouette scores. This analysis segregated the patients into two clusters, indicating two distinct immune response profiles among the bronchiectasis patients in our study. Subsequent examination of biomarker expression demonstrated statistically significant differences in 14 out of the 16 biomarkers between the clusters. The biomarkers that show significant alterations include CD62E, IFNα, IFNγ, IL-1α, IL-1β, IL-12p70, IL-13, IL-17A, IL-4, MCP-1, MIP-1α, and TNFα. There were near-significant trends for ICAM-1 (*P* values = 0.059) and IP-10 (*P* value = 0.061). MIP-1β and CD62P did not show any significant changes. Cluster analysis indicated that one cluster exhibited consistently lower levels of all significantly altered biomolecules (Fig. 2A). Further validation through principal component analysis (PCA) distinctly segregated the clusters into low-level and high-level cytokines and chemokines production groups, reinforcing the clustering outcomes (Fig. 2B).

The first cluster, which we named the low inflammatory response group (LRG), includes 27 individuals with relatively low levels of anti-inflammatory and pro-inflammatory cytokines and chemokines. This pattern suggests a relatively low immune

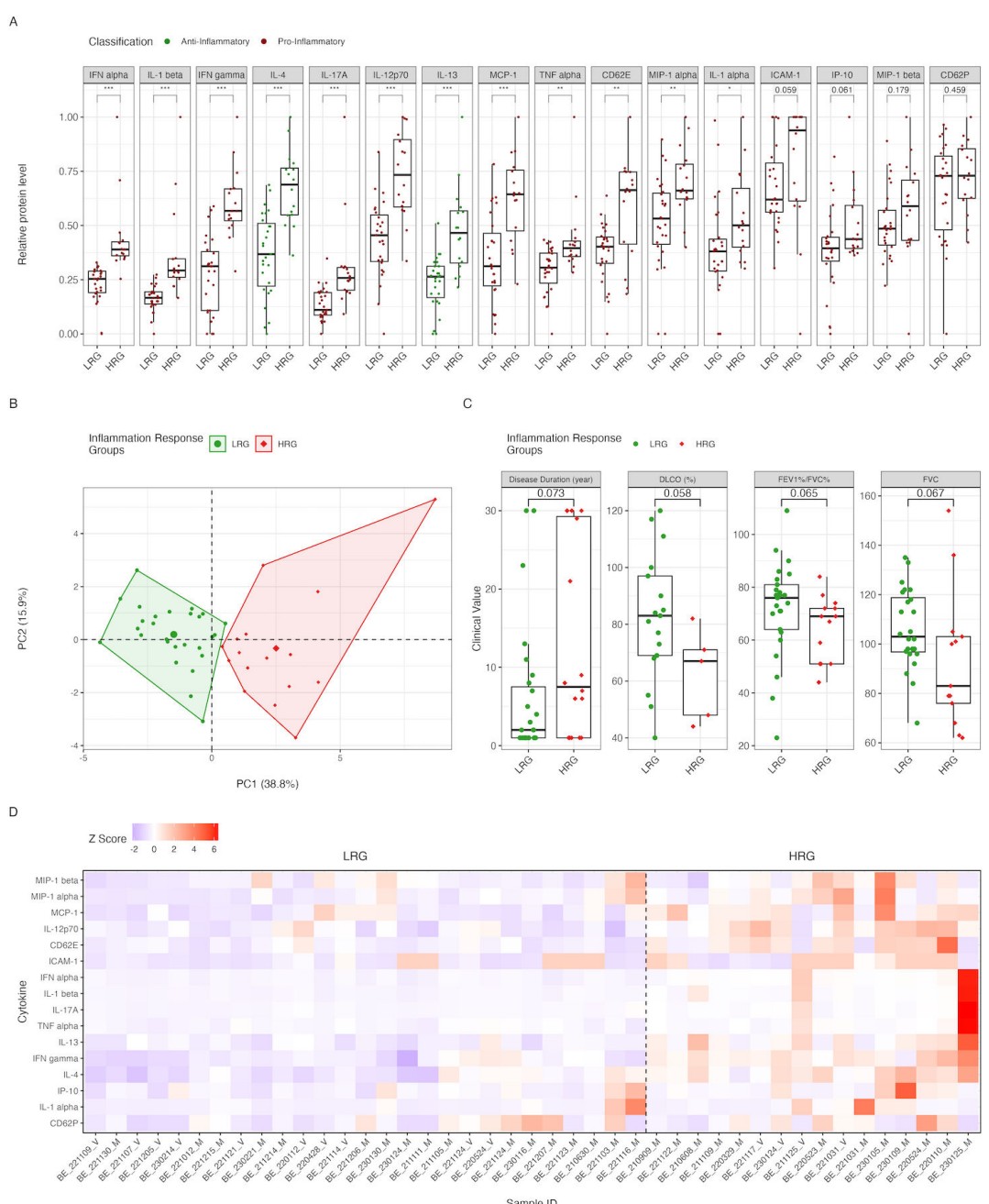

**FIG 2** Analysis of inflammatory response and clinical metrics. (A) Box plots display the relative protein levels of various inflammatory markers between the LRG and HRG. Provided cytokines and chemokines classification based on their role in inflammation: pro-inflammatory (red) and anti-inflammatory (green). Significant differences were observed in several markers, including IFNα, IL-1β, IFNγ, IL-4, IL-17A, IL-12p70, IL-13, MCP-1, TNFα, CD62E, and MIP-1α, all of which were higher in the HRG ($P < 0.001$). IL-1α and ICAM-1 were also significantly higher in the HRG ($P < 0.01$ and $P < 0.05$, respectively). Markers such as IP-10, MIP-1β, and CD62P showed no significant differences between the groups ($P = 0.059$, $P = 0.179$, and $P = 0.459$, respectively). (B) The PCA plot distinguishes between the LRG and HRG based on the inflammatory response, with PC1 (38.8% variance) and PC2 (15.3% variance). The LRG (green) and HRG (red) are clearly separated, indicating significant differences in inflammatory protein expression profiles. (C) Box plots compare clinical metrics between the LRG (green) and HRG (red). (D) The heatmap visualizing relative serum concentrations of inflammatory mediators with color intensity indicating protein level. Significance is indicated as *$P < 0.05$, **$P < 0.01$, ***$P < 0.001$.

response, indicating a clinically less active bronchiectasis or the effect of effective disease management.

Another cluster, which we have named the high inflammatory response group (HRG), displays a pattern indicative of an active immune response with simultaneous upregulation of both pro- and anti-inflammatory molecules. This pattern denotes an active immune response with the concurrent upregulation of pro- and anti-inflammatory molecules, signifying a complex and balanced inflammatory condition, potentially representing an exacerbated stage of the disease. A complex cytokine and chemokine upregulation was demonstrated, reviewed, and discussed in the context of several chronic inflammatory diseases (41). Certain cytokines may become targets for therapy; however, the benefits of such therapy are not clear, although blocking certain individual cytokines has been effective in some chronic inflammatory diseases.

Upon evaluating various clinical, demographic factors, and respiratory function outcomes, we did not find any statistically significant correlations ($P$ value < 0.05) between the inflammatory response groups. However, several strong trends were observed. Notably, HRG patients had a longer history of BE, with a $P$ value of 0.073, compared to LRG (Fig. 2C). This suggests a potential association between prolonged disease duration and upregulated inflammatory responses. Additionally, patients in the HRG exhibited a significantly higher number of damaged lung segments, indicating a more advanced stage of bronchiectasis in these individuals.

Furthermore, in terms of pulmonary function, the HRG demonstrated significantly lower DLCO values, consistent with the observed increased lung damage. This reduction in DLCO ($P$ value = 0.06), a measure of the lung's capacity to transfer gas, often indicates extensive lung damage. Additionally, the HRG showed a notably lower FEV1%/FVC% ratio ($P$ value = 0.064) and FVC ($P$ value = 0.67) compared to the LRG, highlighting a more pronounced airflow obstruction in HRG patients (Fig. 2C).

## The relationship between inflammatory groups and microbiome composition in bronchiectasis

In our study, we explored the relationship between inflammatory response groups and microbiome composition in bronchiectasis patients. Initially, we compared the alpha diversity between the LRG and HRG. Still, we found no significant differences (Fig. 3A). Both groups included patients with high-alpha diversity (Shannon index > 1) and those with low-alpha diversity (Shannon index < 1). Among LRG patients, 70% (19 out of 27) showed high diversity, whereas 30% (8 out of 27) showed low diversity. In contrast, a larger proportion of HRG patients exhibited low-alpha diversity, with 56% (9 out of 16) showing low diversity and 44% (7 out of 16) showing high diversity.

We also examined differences in microbiome composition between the inflammatory response groups. We found significant differences in the taxonomic classes *Betaproteobacteria* and *Negativicutes*, with an enrichment of bacteria from these classes in the LRG (Fig. 3B). To further investigate, we separated the inflammatory groups based on alpha diversity to identify statistically significant differences in microbiome composition. By comparing the LRG low-alpha-diversity subgroup with the HRG low-alpha-diversity subgroup, we identified an enrichment of bacteria from the genera *Gemella*, *Neisseria*, *Prevotella*, *Selenomonas*, and *Veillonella* in the LRG (Fig. 3C). In contrast, we did not find any statistically significant enrichment of bacteria in any taxonomic class in the HRG . Furthermore, we did not find statistical differences in high-alpha-diversity subgroups between the LRG and HRG.

All patients with low-alpha diversity were from monospecies-dominated microbiome clusters 9, 10, 11, 12, 14, and 15 (Fig. 1 and 3D). We found that patients in the LRG with low-alpha diversity were associated with specific dominant bacteria: *H. influenzae* in microbiome cluster 9, one patient in microbiome cluster 10 with *P. stutzeri* as the dominant bacteria, one patient in microbiome cluster 11 with *E. coli* as the dominant bacteria, and patients in microbiome cluster 15 with *K. pneumoniae* dominance.

Crucially, in the low-alpha-diversity subgroup of the LRG , we identified a higher presence of bacteria common to healthy lungs compared to the low-alpha-diversity subgroup of the HRG , as mentioned earlier. Although the relative abundance of these

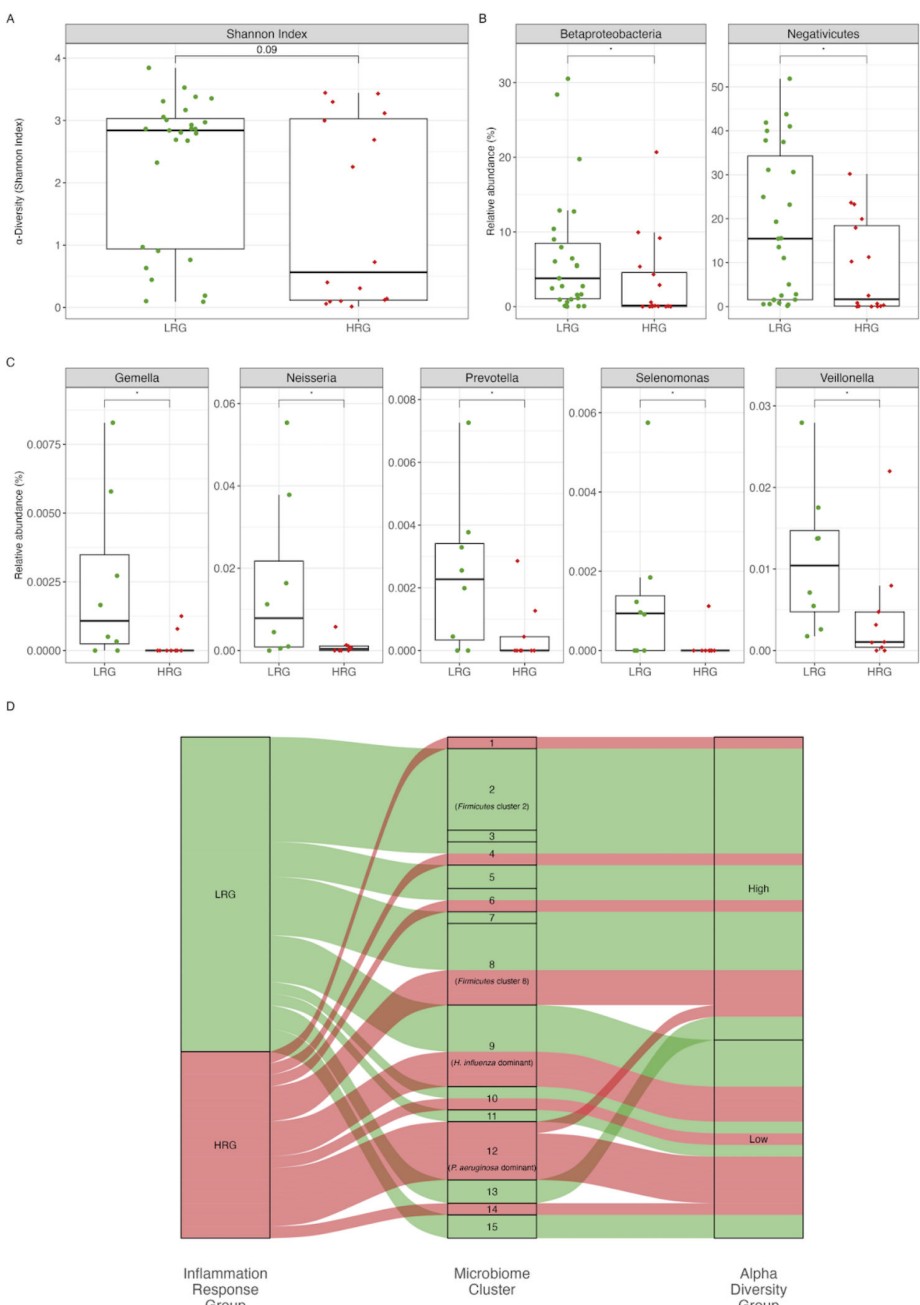

**FIG 3** Microbiome analysis and inflammatory response. (A) Box plots show the alpha diversity (Shannon index) of the microbiome between the LRG and HRG. The Shannon index is slightly higher in LRG compared to HRG ($P = 0.09$), indicating a trend toward greater microbial diversity in LRG. (B) Box plots display the relative abundance of *Betaproteobacteria* and *Negativicutes* between LRG and HRG. *Betaproteobacteria* are significantly more abundant in LRG ($P < 0.05$), whereas *Negativicutes* are significantly more abundant in HRG ($P < 0.05$). (C) Box plots show the relative abundance of specific bacterial genera between LRG and HRG. Genera such as *Gemella*, *Neisseria*, *Prevotella*, *Selenomonas*, and *Veillonella* are significantly more abundant in LRG than in HRG ($P < 0.05$ for all). (D) The Sankey diagram provides a visual representation of the relationship between inflammatory response groups (LRG and HRG), microbiome clusters, and alpha-diversity groups (Shannon index >1 in high-alpha-diversity group; Shannon index <1 in low-alpha-diversity group). It effectively shows the distribution of LRG and HRG across different microbiome clusters, highlighting the connections between specific clusters and overall microbial diversity. Significance is indicated as *$P < 0.05$.

bacteria is significantly lower than in the high-alpha-diversity subgroups, this is possibly why we do not see a high immune response in the LRG . This finding presents intriguing possibilities for further research. By employing more quantitative approaches, we could explore how the loss of common microbiome composition in the lungs of bronchiectasis patients might be linked to the chronic inflammation observed in HRG patients with low-alpha diversity. These insights could significantly advance our understanding of bronchiectasis and foster the development of more effective treatment strategies.

## Microbiome-driven inflammatory marker variations

We also analyzed the relationship between previously defined microbiome clusters and inflammatory response groups. We looked only at these microbiome clusters containing at least three samples to explore. These clusters include *Firmicutes* clusters 2 and 8, *H. influenzae*-dominant, and *P. aeruginosa*-dominant clusters. As shown in Figure 3B, most microbiome clusters fall into single, specific inflammatory response groups, i.e., 1, 3, 6, 7, 11–14, suggesting that certain microbiome signatures correlate with the cytokine/chemokine signatures.

When analyzing specific inflammatory mediators' levels in the context of microbial composition in the lower airways (Fig. 4), additional microbiome divisions were established, including *P. aeruginosa*-dominant, *Firmicutes*-dominant, and *H. influenzae*-dominant clusters. CD62E (E-selectin) concentrations were markedly elevated in the *P. aeruginosa*-dominant cluster, distinguishing it significantly from the *Firmicutes*- and *H. influenzae*-dominant clusters. This distinction was similar to the levels of IL-12. Additionally, IL-1β and IL-17A, associated with Th17 responses, were detected in higher levels in the *P. aeruginosa*-dominant cluster compared to the *Firmicutes* cluster. The chemokines MIP-1α and MIP-1β were upregulated in monospecies-dominated clusters (*H. influenzae* and *P. aeruginosa* dominated) compared to the *Firmicutes* supercluster. MIP-1α demonstrated statistically significant variations between the clusters, whereas MIP-1β, although not reaching a conventional statistical significance with a $P$ value of 0.069, displayed a similar trend (Fig. 4). Notably, the differences in MIP-1 chemokine levels between *H. influenzae*- and *P. aeruginosa*-dominant clusters were not statistically significant. These results prove the complex correlation between certain microbiome clusters and inflammatory response signatures.

## DISCUSSION

The composition of a healthy lower respiratory tract microbiome depends on the passage of the microorganisms from the upper respiratory tract, oral cavity, and gut, their ability to persist and multiply in the lower respiratory tract, and their elimination (42). Bacteria from both oral and nasal cavities can be introduced to the lungs; however, it has been previously observed that the healthy lower respiratory tract microbiome is more similar to the oral microbiota and most similar to the one found in the oropharynx (43–45). Thus, the presence of oral bacteria, such as *Prevotella, Streptococcus,* and *Veillonella*, can be expected and are found in healthy lungs. As *Haemophilus* and *Neisseria* species can also be found in the healthy upper respiratory tract, their presence in the lungs does not always indicate disease (44, 46).

In this study, when the lower respiratory tract microbiomes of BE patients were grouped according to their similarity, the two major types of microbiota were observed: the higher-diversity group (*Firmicutes* clusters, 1–8) and the lower-diversity group, displaying a single species of dominant bacteria. The higher-diversity group displayed a variety of *Firmicutes*, mostly belonging to *Veillonella* and *Streptococcus* genera, indicating a microbiome similarity to a possibly healthy lung microbiota. In the same *Firmicutes* clusters, we have detected *Haemophilus* species, especially *H. parainfluenzae;* however, these *Proteobacteria* are also commonly detected in healthy people's oropharynx (44, 46). In this patient group, the composition of microbiota cannot be a clear indicator and contributor to BE development. However, the microbiota clusters dominated by one bacterial species indicate a shift away from a healthy microbiome. Similar to our

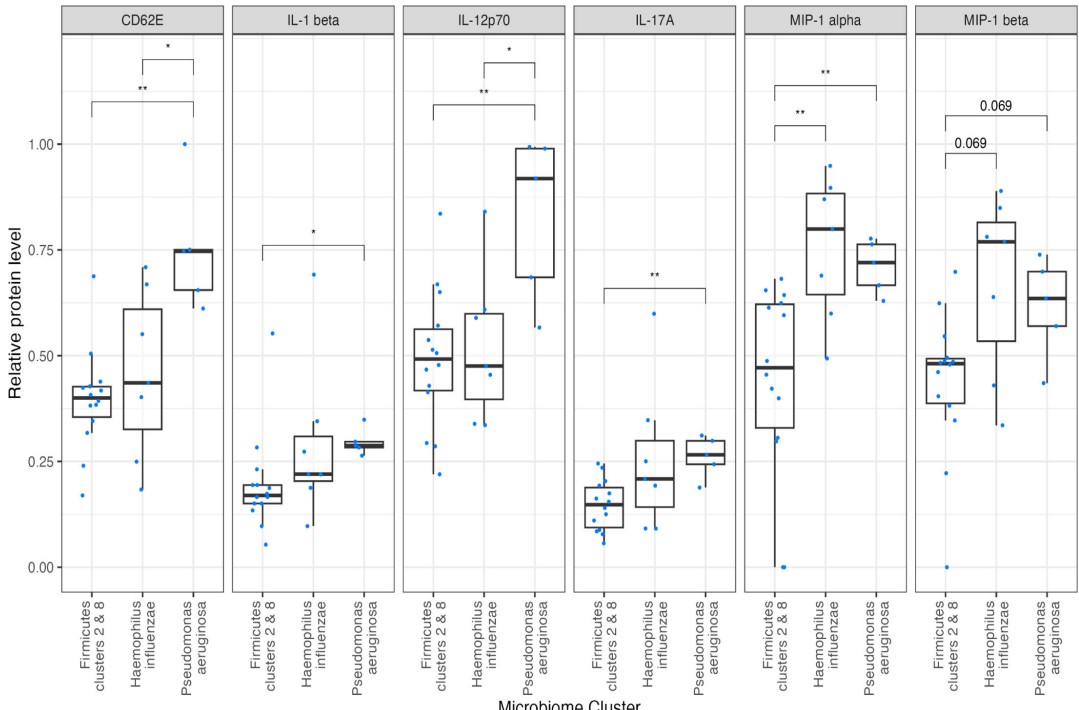

**FIG 4** Differential expression of inflammatory markers across microbiome clusters. Box plots depict the relative abundance (%) of inflammatory markers (CD62E, IL-1β, IL-12p70, IL-17A, MIP-1α, and MIP-1β) across major microbiome clusters: *Firmicutes, Haemophilus influenzae,* and *Pseudomonas aeruginosa*. Asterisks indicate significant differences between clusters: * represents a *P* value < 0.05, and ** denotes a *P* value < 0.01. Notably, MIP-1 beta exhibits a near-significant trend with a *P* value of 0.069.

findings, *H. influenzae* and *P. aeruginosa* have been previously reported as the most common dominant species in the microbiomes of the BE patients' lower airways (47–49) *P. aeruginosa* is commonly associated with the worse outcome of the disease (50, 51). A similar worse outcome has also been observed for the patients that have dominant *Enterobacteriaceae* (49). Bacteria belonging to the *Enterobacteriaceae* family have often been detected in sputum from BE patients using culture analysis (52–54) and are associated with increased severity of disease (49). Together with *P. aeruginosa,* the instances of *Enterobacteriaceae* tend to decrease with repetitive antibiotic treatment (55). However, the diversity also tended to decrease after treatment. The other *Enterobacteriaceae* species have been less common in BE cases. For instance, *K. pneumoniae* has been detected in the sputum of various etiology BE patients via culturing methods in Taiwan's cohort (56); however, it is unclear if they represented a dominant species in the cases. *E. coli* has been isolated from sputum samples of BE and COPD patients; however, it is not a common lung pathogen (57, 58).

Non-aeruginosa *Pseudomonas* species have been previously isolated from pulmonary disease patients (59, 60); however, they were not usually common in BE microbiomes (47). In our study, two patients had a dominant *P. stutzeri*; however, both patients also had a low abundance of *P. aeruginosa* (Table S2). Interestingly, in all *Pseudomonas*-dominant samples, several other non-dominant *Pseudomonas* species were detected, the most common one being not only *P. aeruginosa* and *P. stutzeri,* but also *P. putida*, *P. fluorescence, P. oleovorans,* and others that were found in individual samples. The presence of several *Pseudomonas* species in the lower respiratory tract microbiotas could indicate the shift in the lower airways and lungs environment that is specifically favorable for *Pseudomonas* species.

The largest cluster of these study participants had *H. influenzae* as a dominant microorganism. *H. influenzae* is an obligate human pathogen and may cause severe infectious diseases (affecting meninges, lungs, etc.) in both children and adults. An

effective and safe vaccine against it has been available since the 1980s and was included in national immunization programs. The most invasive *H. influenzae* infections can be prevented with immunization. A recent study shows that the immune mechanism of protection against *H. influenzae* involves Th17 cells (61). In addition, the *Haemophilus*-dominant microbiome tends to persist over time with no effect of treatment (62). Similarly, the microbiomes with dominant *P. aeruginosa* are also stable and persistent despite therapy for many years.

While analyzing serum inflammatory biomarkers, we have identified clusters of subjects with distinct levels of anti-inflammatory and pro-inflammatory systemic cytokines (presented in Fig. 2). The cytokine clusters identified via the consensus elbow method and silhouette score did differ significantly in their cytokine levels (Fig. 2B). Individual patients' serum values of pro- and anti-inflammatory mediators were quantified, and the two distinct response groups were identified, i.e., the LRG with low levels of anti- and pro-inflammatory cytokines representing remission of BE, and the HRG with high levels of both anti- and pro-inflammatory cytokines representing exacerbated BE process (Fig. 2). Further analysis of the groups revealed that they might be distinguished very precisely via principal component analysis, suggesting a possible clinical diagnostic and therapeutic importance of these groups (Fig. 2).

When clinical parameters, bacterial diversity, and bacterial content were assessed for the LRG and HRG , some significant differences were found (as indicated in Fig. 2C and 3). We assessed all microbiome clusters and allocated them to the respective inflammatory response groups and diversity (Fig. 3). Although both immune response groups included patients with high-alpha diversity, there was a tendency of higher diversity among LRG patients in comparison to HRG patients. In addition, the groups were significantly different by their microbiome content, i.e., the *Betaproteobacteria* and *Negativicutes* were both more abundant in LRG patients. These classes of bacteria are usually present in the healthy upper respiratory tract of humans, where their functions may include maintaining immune tolerance, preventing pathogens from colonization, and regulating commensal bacterial communities. When assessing LRG with low diversity, *Gemella*, *Neisseria*, *Prevotella*, *Selenomonas*, and *Veillonella* genera were identified, suggesting the presence of normal human respiratory microflora at identifiable levels. This is contrasted by non-detectable enrichment of bacteria in any taxonomic class in the low-diversity HRG subgroup. In addition, the high-alpha-diversity LRG and HRG subgroups were not different significantly by their content. We should also note that low diversity was exclusively associated with monospecies domination in our study. Moreover, the low-diversity LRG cases were all associated with specific bacteria enrichment, i.e., *H. influenzae*, *P. aeruginosa*, *P. stutzeri*, *E. coli*, or *K. pneumoniae*.

We speculate that maintenance of low inflammatory response requires the presence of regular inhabitants of a healthy human respiratory tract, i.e., low-alpha-diversity group within LRG still contains core microbiome members despite their relatively low abundance and diversity. Subsequently, loss of common microbiome components in the BE-affected lungs might be crucial for the upregulation of the inflammatory response. Thus, future treatment strategies for BE may include microbiome restoring or preserving tactics.

Next, we investigated if the most prominently expressed inflammatory mediators were associated with the specific microbiome clusters. We found elevated E-selectin, IL-1β, IL-17A, and IL-12 in the *P. aeruginosa*-dominant cluster, and the chemokines MIP-1α and MIP-1β were upregulated in *H. influenzae*- and *P. aeruginosa*-dominated clusters as shown in Figure 4. The upregulation of E-selectin mentioned above is crucially important for the inflammatory cell extravasation and transmigration to the damaged tissue, whereas IL-1β, IL-17A, and IL-12 maintain and boost inflammatory cell recruitment, damage to the tissue, activation of the proteases, oxidative stress, and other inflammation-related events. In addition, MIP-1α and MIP-1β mediate the recruitment of more antigen-presenting cells and further increase inflammatory damage.

An interplay of immune response mediators across the selected microbiome clusters is complex. The responses investigated include Th1 and Th2, T regulatory, and Th17 responses based on their serum cytokine content. Th1 utilizes IL-6, -8, -2, -12, and IFN-γ, whereas Th2 cells secrete IL-4, -5, and -10. Th1 cytokines predominantly promote cell-mediated immunity and help in the clearance of intracellular pathogens, whereas Th2 cells are responsible for humoral immunity and are considered anti-inflammatory. The upregulated IL-12 in some of our patients indicates a Th1-type response. IL-1β is a pro-inflammatory cytokine orchestrating inflammation and maintaining the production of other inflammatory molecules, chemokines, antimicrobial peptides, and remodeling proteins. The IL-17A level detected and the associated Th17 response mediate host defense against bacteria and, possibly, an autoimmune component of the disease. The significant upregulation of pro-inflammatory mediators in the *H. influenzae*- and *P. aeruginosa*-dominated microbiome clusters corroborates reports by other groups and illustrates simultaneously developing microbial composition shift and serum inflammatory mediator content increase.

The importance of the unique set of mediators upregulated in *P. aeruginosa*- and *H. influenzae*-dominated microbiome carriers is defined by the function of those mediators. It includes pro-inflammatory regulation, extravasation of immune cells, chemotaxis into the sites of damage, and some others, similar to Th17-mediated protection against *P. aeruginosa*-induced pneumonia (63). It is clear that specific cytokine and chemokine signatures are dictated by individual microbiome content and might become valuable resources for the diagnostic biomarkers and treatment targets in an era of personalized medicine.

## Conclusions

- Lung microbiome diversity in bronchiectasis patients is represented by two types of microbiomes. One group of microbiomes consists of a single dominant bacterial species, such as *H. influenzae, P. aeruginosa,* or bacteria from *Enterobacteriaceae* family, colonizing the patient's lungs. The other group of microbiomes exhibits higher microbial diversity in their lower respiratory tract, resembling the composition of a healthy lung microbiota.
- Systemic immune responses in our cohort were evaluated, and two distinct response groups were identified, particularly low- and high-response groups. Each group had a specific array of clinical symptoms, microbial composition, and diversity. Such divisions represent clinically important bronchiectasis disease subgroups with potentially unique diagnosis and treatment targets.
- Immune response groups match specific bacterial composition in the hosts' lower airways, where *Pseudomonas*- or *Haemophilus*-dominated microbiomes are indeed associated with high inflammatory response, i.e., high levels of pro- and anti-inflammatory cytokines.
- Certain genera-containing microbiomes correlate with low inflammatory responses, suggesting possible probiotic strains, e.g., members of *Gemella*, *Neisseria*, *Prevotella*, *Selenomonas*, and *Veillonella* genera.

In sum, this study opens new insights into microbiome–host interrelationship and provides a valuable resource of candidate biomarkers or therapeutic targets for better bronchiectasis management in the future.

## ACKNOWLEDGMENTS

This work was supported by a grant (#01.2.2-LMT-K-718-03-0079) from the Lithuanian Research Council.

## AUTHOR AFFILIATIONS

[1]Life Sciences Center, Institute of Biosciences, Vilnius University, Vilnius, Lithuania

²State Research Institute Centre for Innovative Medicine, Vilnius, Lithuania

³Clinic of Chest Diseases, Immunology, and Allergology, Faculty of Medicine, Vilnius University, Vilnius, Lithuania

⁴Centre of Pulmonology and Allergology, Vilnius University Hospital Santaros Klinikos, Vilnius, Lithuania

## AUTHOR ORCIDs

Rūta Aldonytė ⓘ http://orcid.org/0000-0002-9806-025X

## FUNDING

| Funder | Grant(s) | Author(s) |
|---|---|---|
| Lithuanian Research Council | 01.2.2-LMT-K-718-03-0079 | Aleksandras Konovalovas |
| | | Julija Armalytė |
| | | Laurita Klimkaitė |
| | | Tomas Liveikis |
| | | Brigita Jonaitytė |
| | | Edvardas Danila |
| | | Daiva Bironaitė |
| | | Diana Mieliauskaitė |
| | | Edvardas Bagdonas |
| | | Rūta Aldonytė |

## ADDITIONAL FILES

The following material is available online.

Open Peer Review

**PEER REVIEW HISTORY (review-history.pdf).** An accounting of the reviewer comments and feedback.

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
