## [Reviewer comments · Microbiology Spectrum]

Microbiology Spectrum

Insights into respiratory microbiome composition and systemic inflammatory biomarkers of bronchiectasis patients

Aleksandras Konovalovas, Julija Armalytė, Laurita Klimkaitė, Tomas Liveikis, Brigita Jonaitytė, Edvardas Danila, Daiva Bironaitė, Diana Mieliauskaitė, Edvardas Bagdonas, and Ruta Aldonyte

Corresponding Author(s): Ruta Aldonyte, State Research Institute Center for Innovative Medicine

Review Timeline:

Submission Date:	December 7, 2023
Editorial Decision:	March 18, 2024
Revision Received:	May 24, 2024
Accepted:	July 17, 2024

Editor: Cecilia Thompson

Reviewer(s): Disclosure of reviewer identity is with reference to reviewer comments included in decision letter(s). The following individuals involved in review of your submission have agreed to reveal their identity: Sutonuka Bhar (Reviewer #1); Danielle Elizabeth Campbell (Reviewer #2); Evann Hilt (Reviewer #3)

Transaction Report:

DOI: <https://doi.org/10.1128/spectrum.04144-23>

Re: Spectrum04144-23 (Insights into respiratory microbiome composition and systemic inflammatory biomarkers of bronchiectasis patients)

Dear Dr. Ruta Aldonyte:

Thank you for the privilege of reviewing your work. Below you will find my comments, instructions from the Spectrum editorial office, and the reviewer comments.

Revision Guidelines

Sincerely,
Cecilia Thompson
Editor
Microbiology Spectrum

Reviewer #1 (Comments for the Author):

The article by Konovalovas et al. correlates different respiratory microbial population abundance with severity of disease, evident by presence of pro-inflammatory and anti-inflammatory cytokines in the samples obtained from bronchiectasis patients. The article is scientifically sound and very well explained with appropriate data and detailed methods and results. I have some minor suggestions for the authors:

Line 30: "including the" should be replaced with "including ones present in the".

Line 55-57: This sentence is vague and some of these "testing technologies" can be specified.

Line 60-62: The studies are called -omics, i.e. the words should end in -omics. For example, metabolomics, exposomics, etc.

Line 183: Under "Analysis of inflammatory serum biomarkers", mention if samples were run in multiple dilutions or if only a single dilution was run. If multiple dilutions were not run, samples at the higher end run the risk of being limited by assay ULOQ and this info should be mentioned in the text.

Line 192: More information on the normalization procedure would be ideal to include.

Line 206-207: Add reasoning/references showing how these two types of samples (bronchial aspirate and BALF), although different, are comparable to each other for purpose of this study.

Line 337 and 499: There is a lot of information in both these results sections, hence, it will be informative to add a couple concluding sentences at the end of each result section.

Line 355: I would challenge that IL-12 is generally considered pro-inflammatory. Kindly provide reference showing it's anti-inflammatory properties in the respiratory system.

Line 360-361: it may be good to have a table/figure distributing the patients into these 3 groups and showing the levels of different pro/anti-inflammatory cytokines in each patient.

Reviewer #3 (Comments for the Author):

The authors of "Insights into respiratory microbiome composition and systemic inflammatory biomarkers of bronchiectasis patients" present a compelling body of work that tackles the question of if there is a connection between respiratory microbiome composition and inflammatory status of this patient group. They have an extensive workup of both the respiratory microbiome and the various inflammatory markers that have been implicated in the disease state of bronchiectasis. I have a few minor edits/suggestions below.

Edits/Comments/Suggestions

Line 103-Change the word "little" to "poorly"

Line 119-131-Should write up the statistics performed on the clinical metrics. Similar to how you have described in sections to follow.

Line 150-Should cite the guidelines provided by the manufacturer

Line 209-213-These two sentences are redundant. Suggested edit to one sentence "A notable characteristic of the studied populations was the substantial variation in disease duration, which averaged around 8.6 years (SD=10.4), underscoring the commitment to examine a wide variety of patients and disease progressions.

Line 278-Should reference Figure 1 after the first sentence of the paragraph.

Line 283-Please name the dominant species of bacteria. And is this one bacteria species for all 4 clusters (9-12) or just one of them?

Line 287-289-Would be helpful to have the cluster number listed here to help the reader connect to the Figure.

Line 291-292- So this organism was cultured and then reported out as a pathogen? Were the patients treated based on the culture results?

Line 299- Suggested edit to match the way it is presented in Line 298: "(20 patients, 42.6%)"

Line 305-307-Again, was the organism reported out as a pathogen? Were the patients treated based on culture results?

Line 324-"cluster groups"-should be plural

Line 415-Typo, should be HRG not HIRG

Line 489-Should reference Figure 6 after the first sentence of the paragraph

Line 524- Second Enterobacteriaceae is spelled incorrectly.

Line 586- Italicize *P. aeruginosa*

Line 618-Italicize Enterobacteriaceae

Konovalovas *et al.*, present an intriguing analysis of the microbiome and inflammatory markers of BE patients. Using whole-gene 16S sequencing, they identify 15 clusters of lung microbiomes in their cohort of 47 BE patients. Simultaneously, they analyze inflammatory marker profiles from serum, identifying 2 pro-inflammatory and 2 anti-inflammatory clusters of individuals, and 3 “response groups” spanning these four clusters. They associate this inflammatory response with a number of disease severity markers, and abundances of specific microbiota. The authors claim that “most microbiome clusters fall into single specific inflammatory response groups,” though this reviewer is unclear how they arrived at this conclusion (see comments below).

A major pitfall of this study is the lack of any healthy control group. For lung microbiome analyses, serum inflammatory biomarkers, as well as clinical disease phenotypes (Fig4A), it is unclear to the reader which markers distinguish BE disease from health. A healthy control group should be included in all analyses.

Most of the analyses presented define groups of individuals/samples, including microbiome clusters, pro- and anti-inflammatory response clusters, and response groups. Though this can be useful for some analyses, this reviewer recommends a more thorough correlative examination of many of the analyses presented. For many of the analyses currently presented appear to potentially mask broader patterns outside of those groups presented. For example, instead of using box plots in Fig 6, would scatter plots better show correlations between specific bacterial abundances and inflammatory marker abundances? These analyses can be extended to all microbiome clusters, rather than restricting the analyses to only the four shown here, and may lead to more broadly applicable conclusions. Likewise, Fig. 5B could be represented as a heatmap for all bacterial taxa in each response group, more similar to Fig 1.

Fig 2: Can this figure be reframed to use LRG, HRG, and DRG on the x-axis? This would make for significantly clearer presentation and continuity of the results.

Fig 5B: The authors state in the results (lines 482-483) that only microbiome clusters with at least 3 samples were analyzed. However, Fig 5B presents all microbiome clusters, including those with apparently only one individual. In lines 484-485 clusters 1, 3, 6, 7, 11, 13, and 14 are highlighted, despite these clusters having apparently fewer than 3 individuals per cluster, according to Fig 1.

Lines 487-499: This discussion of the microbiome clusters is the clearest and, in this reviewer’s opinion, most valuable discussion of the correlation between lung microbiome composition and inflammatory response. Discussing the qualities of the microbiome clusters (e.g., “Haemophilus influenzae-dominant”) earlier in the text when presenting Fig 1 would be very helpful to the reader. However, the presentation of Fig 6 uses microbiome clusters with names that are inconsistent with how they were named previously (e.g., in Fig 1). This reviewer recommends making the naming of microbiome clusters consistent throughout the text and figures.

Response to Reviewers

Reviewer #1

1. Line 30: "including the" should be replaced with "including ones present in the" – *First of all, many thanks for the positive evaluation. This revision will be made as suggested. Please, see the updated manuscript, Line 30.*
2. Line 55-57: This sentence is vague and some of these "testing technologies" can be specified. – *We are introducing a sentence “Recently, microbiota science dramatically excelled due to the progress of emerging sequencing technologies, permitting compositional and functional assessments, and forever shifted our knowledge about the role of microorganisms in human health and disease” to better reflect the situation. Line 72.*
3. Line 60-62: The studies are called -omics, i.e. the words should end in -omics. For example, metabolomics, exposomics, etc. – *We address this by rephrasing the sentence. The current sentence is: “The broader studies, where host/microorganisms interactions are characterized using -omics approach, e.g., metabolomics, exposomics, secretomics and similar, are increasingly performed”. Lines 77-78.*
4. Line 183: Under "Analysis of inflammatory serum biomarkers", mention if samples were run in multiple dilutions or if only a single dilution was run. If multiple dilutions were not run, samples at the higher end run the risk of being limited by assay ULOQ and this info should be mentioned in the text. – *We appreciate the comment. We indicate in the updated version of the manuscript that a single dilution was used (Line 203). In addition, we can confirm that the concentrations measured were mainly in the middle range of the curve.*
5. Line 192: More information on the normalization procedure would be ideal to include. *We extend the sentence: “Protein targets detected in less than 60% of samples were excluded from the analysis” with “this included IL-6 and IL-8; IL-10 was also excluded from the analysis because it showed a detectable signal in only 4 out of the 43 samples. Similarly, GM-CSF was excluded due to detectable signals in only 3 out of 43 samples” Lines 212-214.*

6. Line 206-207: Add reasoning/references showing how these two types of samples (bronchial aspirate and BALF), although different, are comparable to each other for purpose of this study. – *We are adding the ending phrase: “...which are both sample types widely used for microbiome testing”. We have employed both sample types since the literature review and experience reveal that both sample types can be successfully used and compared for microbiome analysis. Line 228..*
7. Line 337 and 499: There is a lot of information in both these results sections, hence, it will be informative to add a couple of concluding sentences at the end of each result section. – *Thank you for this wise suggestion. We have amended the entire Results section significantly. It is, hopefully, shorter and clearer now.*
8. Line 355: I would challenge that IL-12 is generally considered pro-inflammatory. Kindly provide reference showing it's anti-inflammatory properties in the respiratory system. *That is a very correct comment. We agree that IL-12 is pro-inflammatory and thus we are amending all relevant sections in Results and Discussion. Inflammatory response groups are different now. The entire division describing immune mediators is much shorter in the updated version of the manuscript. Attributing IL-12 to pro-inflammatory and IL-10 to anti-inflammatory groups caused the rearrangement of Figures, too. We have new Figures 2 and 3 now, which are produced to replace Figures 2, 3, 4, and 5 in the previous version of the manuscript. Please see the marked-up part of the Results, Lines 350-429.*
9. Line 360-361: it may be good to have a table/figure distributing the patients into these 3 groups and showing the levels of different pro/anti-inflammatory cytokines in each patient.- *We have updated the mentioned segment and now have just two groups of patients. Their cytokine levels (absolute and relative numbers) are now provided in Supplementary Table #3 and also visually in Figure 2 (heatmap).*

Reviewer #3

1. Line 103 - Change the word "little" to "poorly". - *Many thanks for the positive evaluation. The manuscript is now revised as suggested. Please see Line 119.*
2. Line 119-131 - Write up the statistics performed on the clinical metrics, similar to how you have described in the sections to follow. - *Thank you for the question. No statistical comparisons were performed on the clinical metrics themselves. Instead, the clinical data presented in Tables 1 and 2 and Supplementary Table 1 include descriptive statistics to provide a comprehensive overview of the patient cohort. Specifically, the data is summarized as follows: Average, The mean value of each clinical metric; Standard Deviation (SD), The measure of variability around the mean; Median, The middle value of each clinical metric, providing a measure of central tendency; Range (min-max): the minimum and maximum values observed, illustrating the spread of the data. We focused our statistical analysis on patient groups and clusters defined by microbiome and inflammation biomarker data, as described in detail in the article.*
3. Line 150 - Should cite the guidelines provided by the manufacturer. - *We have expanded the section with this inclusion, Lines 167-168: "..., following the protocols provided by Oxford Nanopore Technologies (available at <https://community.nanoporetech.com>)."*
4. Line 209-213 - These two sentences are redundant. Suggested edit to one sentence "A notable characteristic of the studied populations was the substantial variation in disease duration, which averaged around 8.6 years (SD=10.4), underscoring the commitment to examine a wide variety of patients and disease progressions. - *Revised as suggested.*
5. Line 278 - Should reference Figure 1 after the first sentence of the paragraph. - *Revised as suggested, i.e., Figure 1 is referenced at the beginning of the paragraph. Please see the updated manuscript, Line 291.*
6. Line 283 - Please name the dominant species of bacteria. And is this one bacteria species for all 4 clusters (9-12) or just one of them? – *We have rephrased "one" into "a single" to better reflect the situation. Line 296 and elsewhere.*

7. Line 287-289 - It would be helpful to have the cluster number listed here to help the reader connect to the Figure. - *We have added the requested information, i.e., "(clusters 9-15)", in Line 299.*
8. Line 291-292—So, this organism was cultured and then reported out as a pathogen? Were the patients treated based on the culture results? - *Correct, the treatment was administered based on clinical indications and culture results.*
9. Line 299 - Suggested edit to match the way it is presented in Line 298: "(20 patients, 42.6%)" – *Thank you. Edited as suggested. Line 312.*
10. Line 305-307 - Again, was the organism reported out as a pathogen? Were the patients treated based on culture results? - *Correct, the treatment was administered based on clinical indications and culture results.*
11. Line 324 - "cluster groups" - should be plural – *Corrected, Line 336.*
12. Line 415 - Typo, should be HRG not HIRG – *Section was rewritten.*
13. Line 489 - Should reference Figure 6 after the first sentence of the paragraph - *Corrected, Line 438. It is Figure 4 in the updated version of the manuscript.*
14. Line 524 - Second Enterobacteriaceae is spelled incorrectly. - *Thank you. Corrected, Line 476.*
15. Line 586 - Italicize *P. aeruginosa* – *Corrected.*
16. Line 618 - Italicize Enterobacteriaceae - *Corrected, Line 568.*

Re: Spectrum04144-23R1 (Insights into respiratory microbiome composition and systemic inflammatory biomarkers of bronchiectasis patients)

Dear Dr. Ruta Aldonyte:

Your manuscript has been accepted, and I am forwarding it to the ASM production staff for publication. Your paper will first be checked to make sure all elements meet the technical requirements. ASM staff will contact you if anything needs to be revised before copyediting and production can begin. Otherwise, you will be notified when your proofs are ready to be viewed.

Sincerely,
Cecilia Thompson
Editor
Microbiology Spectrum

Reviewer #1 (Comments for the Author):

The authors have modified the article and changed their study design and analysis based on the major issues which were pointed out in their first draft of the article. I believe they have addressed all the comments satisfactorily.

Reviewer #3 (Comments for the Author):

All of my previous edits/suggestions have been addressed adequately.

Thank you for the updated manuscript, which has improved. There remains a significant gap in the study, however, which is the lack of a healthy control group of any kind. Further, the data appear to be incompletely analyzed while using group-based comparisons without looking at continuous correlations. As such, this reviewer unfortunately recommends the publication be rejected.

More discussion should be devoted to distinguishing disease from health before analyzing the differences between the two response groups that are delineated, especially for inflammatory markers. Lines 377-387 discuss “low levels” of these markers in LRG individuals, though no comparison of disease vs health is reported. Likewise, the discussion of HRG markers alludes to some complex analysis that defines an inflammatory response, though none is reported in the methods. Even if the authors are unable to obtain new samples from a healthy cohort, they could use control data from previously published analyses.

Most of the analyses presented define groups of individuals/samples, including microbiome clusters, pro- and anti-inflammatory response clusters, and response groups. Though this can be useful for some analyses, this reviewer strongly recommends performing correlative examination of the data outside of groups. The clustered approach of these analyses as currently presented may mask broader patterns outside of those groups presented. For example, instead of using box plots in Fig 4, would scatter plots better show correlations between specific bacterial abundances and inflammatory marker abundances? These analyses can be extended to all microbiome clusters, rather than restricting the analyses to only those shown, and may lead to more broadly applicable conclusions. Even a lack of correlation would be biologically informative for those which remain significant at the level of groups, and would suggest some other metric exists within your defined groups to drive these relationships.

Discussing the qualities of the microbiome clusters (e.g., “Haemophilus influenzae-dominant”) earlier in the text and using them as labels in Fig 1 would be very helpful to the reader. Figure 3 shows both cluster numbers and only some qualitative descriptors, while Fig 4 shows qualitative descriptors, and only some cluster numbers. Consistency and using the clearest names for the microbiome clusters would be appreciated.

Fig 3B: It is confusing that the authors state on lines 442-443 that only microbiome clusters with at least 3 samples were analyzed and then refer to Fig 3B, which presents all microbiome clusters, as well as highlight clusters 1, 3, 6, 7, and 11-14 on line 445, despite these clusters having fewer than 3 individuals per cluster. Try to rephrase this so that it is clear that this 3 sample minimum is only applicable to Fig 4.

Table 1: It is unclear if these statistics apply to all patients or only females. Why are males not shown?

Fig 2B: What does the size of each node indicate?

Fig 4D: This is another instance of needing a correlative graph to better show the pattern you are claiming. Plotting alpha diversity versus a marker of inflammatory response, multiple markers, or some aggregated inflammatory score would better analyze these data. The presentation of this figure as it currently is shown is confusing and likely masks broader patterns.